# iRDA Method
# for Sparse Convolutional Neural Networks

## Abstract

We propose a new approach, known as the iterative regularized dual averaging (iRDA), to improve the efficiency of convolutional neural networks (CNN) by significantly reducing the redundancy of the model without reducing its accuracy. The method has been tested for various data sets, and proven to be significantly more efficient than most existing compressing techniques in the deep learning literature. For many popular data sets such as MNIST and CIFAR-10, more than 95% of the weights can be zeroed out without losing accuracy. In particular, we are able to make ResNet18 with 95% sparsity to have an accuracy that is comparable to that of a much larger model ResNet50 with the best 60% sparsity as reported in the literature.

## 1 Introduction

In recent decades, deep neural network models have achieved unprecedented success and state-of-the-art performance in various tasks of machine learning or artificial intelligence, such as computer vision, natural language processing and reinforcement learning Lecun et al. (2015). Deep learning models usually involve a huge number of parameters to fit variant kinds of datasets, and the number of data may be much less than the amount of parameters He et al. (2016). This may implicate that deep learning models have too much redundancy. This can be validated by the literatures from the general pruning methods Pratt (1988) to the compressing models Han et al. (2015a).

While compressed sensing techniques have been successfully applied in many other problems, few reports could be found in the literature for their application in deep learning. The idea of sparsifying machine learning models has attracted much attention in the last ten years in machine learning Donoho (2006); Xiao (2010). When considering the memory and computing cost for some certain applications such as Apps in mobile, the sparsity of parameters plays a very important role in model compression Han et al. (2015a); Cheng et al. (2017). The topic of computing sparse neural networks can be included in the bigger topic on the compression of neural networks, which usually further involves the speedup of computing the compressed models.

There are many sparse methods in machine learning models such as FOBOS method Duchi and Singer (2009), also known as proximal stochastic gradient descent (prox-SGD) methods Mine and Fukushima (1981), proposed for general regularized convex optimization problem, where $\ell_1$ is a common regularization term. One drawback of prox-SGD is that the thresholding parameters will decay in the training process, which results in unsatisfactory sparsity Xiao (2010). Apart from that, the regularized dual averaging (RDA) method Xiao (2010), proposed to obtain better sparsity, has been proven to be convergent with specific parameters in convex optimization problem, but has not been applied in deep learning fields.

In this paper, we analyze the relation between simple dual averaging (SDA) method Nesterov (2009) and the stochastic gradient descent (SGD) method Robbins and Monro (1951), as well as the relation between SDA and RDA. It is well-known that SGD and its variants work quite well in deep learning problems. However, there are few literatures in applying pure training algorithms to deep CNNs for model sparsification. We propose an iterative RDA (iRDA) method for training sparse CNN models, and prove the convergence under convex conditions. Numerically, we compare prox-SGD with iRDA, where the latter can achieve better sparsity results while keeping satisfactory accuracy on MNIST, CIFAR-10 and CIFAR-100. We also show iRDA works for different CNN models such

as VGG Simonyan and Zisserman (2014) and He et al. (2016). Finally, we compare the performance of iRDA with some other state-of-the-art compression methods.

## 2 RELATED WORKS

Cheng et al. (2017) reviews the work on compressing neural network models, and categorizes the related methods into four schemes: parameter pruning and sharing, low-rank factorization, transfered/compact convolutional filters and knowledge distillation. Among them, Liu et al. (2015) uses sparse decomposition on the convolutional filters to get sparse neural networks, which could be classified to the second scheme. Apart from that, Han et al. (2015b) prunes redundant connections by learning only the important parts. Louizos et al. (2017) starts from a Bayesian point of view, and removes large parts of the network through sparsity inducing priors. Yin et al. (2018) He et al. (2018) combines reinforcement learning methods to compression. Li and Hao (2018) considers deep learning as a discrete-time optimal control problem, and obtains sparse weights on ternary networks. Recently, Feng (2018) applies RDA to fully-connected neural network models on MNIST.

## 3 ALGORITHMS

Let $z = (x, y)$ be an input-output pair of data, such as a picture and its corresponding label in a classification problem, and $f(w, z)$ be the loss function of neural networks, i.e. a scalar function that is differentiable w.r.t. weights $w$. We are interested in the expected risk minimization problem

$$\min_{w} \quad \{\mathbf{E}_z f(w, z)\}. \tag{1}$$

The empirical risk minimization

$$\min_{w} \quad \left\{\frac{1}{T} \sum_{t=1}^{T} f(w, z_t)\right\} \tag{2}$$

is an approximation of (1) based on some finite given samples $\{z_1, z_2, \ldots, z_T\}$, where $T$ is the size of the sample set.

Regularization is a useful technique in deep learning. In general, the regularized expected risk minimization has the form

$$\min_{w} \quad \{\phi(w) = \mathbf{E}_z f(w, z) + \Psi(w)\}, \tag{3}$$

where $\Psi(w)$ is a regularization term with certain effect. For example, $\Psi(w) = \|w\|_2^2$ may improve the generalization ability, and an $\ell_1$-norm of $w$ can give sparse solutions. The corresponding regularized empirical risk minimization we concern takes the form

$$\min_{w} \quad \left\{\phi(w) = \frac{1}{T} \sum_{t=1}^{T} f(w, z_t) + \Psi(w)\right\}. \tag{4}$$

SDA method is a special case of primal-dual subgradient method first proposed in Nesterov (2009). Xiao (2010) proposes RDA for online convex and stochastic optimization. RDA not only keeps the same convergence rate as Prox-SGD, but also achieves more sparsity in practice.

In next sections, we will discuss the connections between SDA and SGD, as well as RDA and Prox-SGD. We then propose iRDA for $\ell_1$ regularized problem of deep neural networks.

### 3.1 SIMPLE DUAL AVERAGING METHOD

As a solution of (2), SDA takes the form

$$w_{t+1} = \arg\min_{w} \left\{\frac{1}{t} \sum_{\tau=1}^{t} \langle g_\tau(w_\tau), w \rangle + \frac{\beta_t}{t} h(w)\right\}. \tag{5}$$

The first term $\sum_{\tau=1}^{t} \langle g_\tau, w \rangle$ is a linear function obtained by averaging all previous stochastic gradient. $g_t$ is the subgradient of $f_t$. The second term $h(w)$ is a strongly convex function, and

$\{\beta_t\}$ is a nonnegative and nondecreasing sequence which determines the convergence rate. As $g_\tau(w_\tau), \tau = 1, \ldots, t - 1$ is constant in current iteration, we use $g_\tau$ instead for simplicity in the following. Since subproblem equation 5 is strongly convex, it has a unique optimal solution $w_{t+1}$.

Let $w_0$ be the initial point, and $h(w) = \frac{1}{2}\|w - w_0\|_2^2$, the iteration scheme of SDA can be written as

$$w_{t+1} = w_0 - \frac{1}{\beta_t} \sum_{\tau=1}^{t} g_\tau = w_0 - \frac{t}{\beta_t} \bar{g}_t, \tag{6}$$

where $\bar{g}_t = \frac{1}{t} \sum_{\tau=1}^{t} \langle g_\tau, w \rangle$. Let $\beta_t = \gamma t^\alpha$, SDA can be rewritten recursively as

$$
\begin{aligned}
w_{t+1} &= w_0 - \frac{1}{\gamma t^\alpha} \sum_{\tau=1}^{t} g_\tau \\
&= w_0 - \frac{1}{\gamma t^\alpha} \left( \frac{(t-1)^\alpha}{(t-1)^\alpha} \sum_{\tau=1}^{t-1} g_\tau + g_t \right) \\
&= \left( 1 - \frac{(t-1)^\alpha}{t^\alpha} \right) w_0 + \frac{(t-1)^\alpha}{t^\alpha} \left( w_0 - \frac{1}{\gamma(t-1)^\alpha} \sum_{\tau=1}^{t} g_\tau \right) - \frac{1}{\gamma t^\alpha} g_t \\
&= \left( 1 - \left( 1 - \frac{1}{t} \right)^\alpha \right) w_0 + \left( 1 - \frac{1}{t} \right)^\alpha w_t - \frac{1}{\gamma t^\alpha} g_t,
\end{aligned}
\tag{7}
$$

where $\left( 1 - \left( 1 - \frac{1}{t} \right)^\alpha \right) \to 0$ and $\left( 1 - \frac{1}{t} \right)^\alpha \to 1$ as $t \to \infty$. Thus, SDA can be viewed as a perturbation of SGD.

### 3.2 PROXIMAL STOCHASTIC GRADIENT DESCENT AND REGULARIZED DUAL AVERAGING METHODS

For the regularized problem (4), we recall the well-known Prox-SGD and RDA method first. At each iteration, Prox-SGD solves the subproblem

$$w_{t+1} = \arg\min_w \left\{ \langle g_t, w \rangle + \frac{1}{2\alpha_t} \|w - w_t\|_2^2 + \Psi(w) \right\}. \tag{8}$$

Specifically, $\alpha_t = \frac{1}{\gamma\sqrt{t}}$ obtains the best convergence rate. The first two terms are an approximation of the original objective function. Note that without the regularization term $\Psi$, equation 8 is equivalent to SGD. It can be written in forward-backward splitting (FOBOS) scheme

$$w_{t+\frac{1}{2}} = w_t - \alpha_t g_t, \tag{9}$$

$$w_{t+1} = \arg\min_w \left\{ \frac{1}{2} \|w - w_{t+\frac{1}{2}}\|_2^2 + \alpha_t \Psi(w) \right\}, \tag{10}$$

where the forward step is equivalent to SGD, and the backward step is a soft-thresholding operator working on $w_{t+\frac{1}{2}}$ with the soft-thresholding parameter $\alpha_t$.

Different from Prox-SGD, each iteration of RDA takes the form

$$w_{t+1} = \arg\min_w \left\{ \frac{1}{t} \sum_{\tau=1}^{t} \langle g_\tau, w \rangle + \Psi(w) + \frac{\beta_t}{t} h(w) \right\}. \tag{11}$$

Similarly, taking $h(w) = \frac{1}{2}\|w - w_0\|_2^2$, RDA can be written as

$$w_{t+1} = \arg\min_w \left\{ \langle \bar{g}_t, w \rangle + \frac{\beta_t}{2t} \|w - w_0\|_2^2 + \Psi(w) \right\} \tag{12}$$

$$= \arg\min_w \left\{ \frac{1}{2} \|w - (w_0 + \frac{t}{\beta_t} \bar{g}_t)\|_2^2 + \frac{t}{\beta_t} \Psi(w) \right\}, \tag{13}$$

or equivalently,

$$w_{t+\frac{1}{2}} = w_0 - \frac{t}{\beta_t}\bar{g}_t, \tag{14}$$

$$w_{t+1} = \arg\min_w \left\{ \frac{1}{2}\|w - w_{t+\frac{1}{2}}\|_2^2 + \frac{t}{\beta_t}\Psi(w) \right\}, \tag{15}$$

where $\beta_t = \gamma\sqrt{t}$ to obtain the best convergence rate. From equation 14, one can see that the forward step is actually SDA and the backward step is the soft-thresholding operator, with the parameter $t/\beta_t$.

### 3.3   $\ell_1$ REGULARIZATION AND THE SPARSITY

Set $\Psi(w) = \lambda\|w\|_1$. The problem (4) then becomes

$$\min_w \sum_{t=1}^{T} f_t(w) + \lambda\|w\|_1, \tag{16}$$

where $\lambda$ is a hyper-parameter that determines sparsity.

In this case, from Xiao's analysis of RDA Xiao (2010), the expected cost $\mathbf{E}\phi(\bar{w}_t) - \phi^\star$ associated with the random variable $\bar{w}_t$ converges with rate $O(\frac{1}{\sqrt{t}})$ when $\beta_t = \gamma\sqrt{t}$. This convergence rate is consistent with FOBOS Duchi and Singer (2009). However, both results assume $f$ to be a convex function, which can not be guaranteed in deep learning. Nevertheless, we can still verify that RDA is a powerful sparse optimization method for deep neural networks.

We conclude the closed form solutions of Prox-SGD and RDA for equation 16 as follows.

1. The subproblem of Prox-SGD

$$w_{t+1} = \arg\min_w \left\{ g_t^T w + \frac{1}{2\alpha_t}\|w - w_t\|_2^2 + \lambda\|w\|_1 \right\} \tag{17}$$

has the closed form solution

$$w_{t+1}^{(i)} = \begin{cases} w_t^{(i)} - \alpha_t(g_t^{(i)} + \lambda), & w_t^{(i)} - \alpha_t g_t^{(i)} > \alpha_t\lambda, \\ 0, & |w_t^{(i)} - \alpha_t g_t^{(i)}| \le \alpha_t\lambda, \\ w_t^{(i)} - \alpha_t(g_t^{(i)} - \lambda), & w_t^{(i)} - \alpha_t g_t^{(i)} < -\alpha_t\lambda. \end{cases} \tag{18}$$

2. The subproblem of RDA

$$w_{t+1} = \arg\min_w \left\{ \bar{g}_t^T w + \frac{\beta_t}{2t}\|w - w_0\|_2^2 + \lambda\|w\|_1 \right\} \tag{19}$$

has the closed form solution

$$w_{t+1}^{(i)} = \begin{cases} w_0^{(i)} - \frac{t}{\beta_t}(\bar{g}_t^{(i)} + \lambda), & w_0^{(i)} - \frac{t}{\beta_t}\bar{g}_t^{(i)} > \frac{t}{\beta_t}\lambda, \\ 0, & |w_0^{(i)} - \frac{t}{\beta_t}\bar{g}_t^{(i)}| \le \frac{t}{\beta_t}\lambda, \\ w_0^{(i)} - \frac{t}{\beta_t}(\bar{g}_t^{(i)} - \lambda), & w_0^{(i)} - \frac{t}{\beta_t}\bar{g}_t^{(i)} < -\frac{t}{\beta_t}\lambda. \end{cases} \tag{20}$$

3. The $\sqrt{t}$-proximal stochastic gradient method has the form

$$w_{t+\frac{1}{2}} = w_t - \alpha_t g_t,$$
$$w_{t+1} = \arg\min_w \left\{ \frac{1}{2}\|w - w_{t+\frac{1}{2}}\|_2^2 + \frac{t}{\beta_t}\Psi(w) \right\}. \tag{21}$$

The difference between $\sqrt{t}$-Prox-SGD and Prox-SGD is the soft-thresholding parameter chosen to be $\sqrt{t}$. It has the closed form solution

$$w_{t+1}^{(i)} = \begin{cases} w_t^{(i)} - \alpha_t g_t^{(i)} - \frac{t}{\beta_t}\lambda, & w_t^{(i)} - \alpha_t g_t^{(i)} > \frac{t}{\beta_t}\lambda, \\ 0, & |w_t^{(i)} - \alpha_t g_t^{(i)}| \le \frac{t}{\beta_t}\lambda, \\ w_t^{(i)} - \alpha_t g_t^{(i)} + \frac{t}{\beta_t}\lambda, & w_t^{(i)} - \alpha_t g_t^{(i)} < -\frac{t}{\beta_t}\lambda. \end{cases} \tag{22}$$

It is equivalent to

$$w_{t+1} = \arg\min_w \left\{ g_t^T w + \frac{1}{2\alpha_t} \|w - w_t\|_2^2 + \frac{\lambda t}{\alpha_t \beta_t} \|w\|_1 \right\}, \tag{23}$$

where the objective function is actually an approximation of

$$\sum_{i=1}^{T} f_i(w) + \frac{\lambda t}{\alpha_t \beta_t} \|w\|_1. \tag{24}$$

We can easily conclude that this iteration will converge to $w = 0$ if $\alpha_t = \frac{1}{\gamma\sqrt{t}}$ and $\beta_t = \gamma\sqrt{t}$.

Now compare the threshold $\lambda_{PG} = \alpha_t \lambda$ of PG and the threshold $\lambda_{RDA} = \frac{t}{\beta_t}\lambda$ of RDA. With $\alpha_t = \frac{1}{\gamma\sqrt{t}}$ and $\beta_t = \gamma\sqrt{t}$, we have $\lambda_{PG} \to 0$ and $\lambda_{RDA} \to \infty$ as $t \to 0$. It is clear that RDA uses a much more aggressive threshold, which guarantees to generate significantly more sparse solutions.

### 3.4    Iterative RDA Method for Deep Neural Networks

Note that when $\Psi = \lambda\|w\|_1$, RDA requires $w_1 = w_0 = 0$. However, this will make deep neural network a constant function, with which the parameters can be very hard to update. Thus, in Algorithm 1, we modify the RDA method as Step 1, where $w_1$ can be chosen not equal to 0, and add an extra Step 2 to improve the performance. We also prove the convergence rate of Step 1 for convex problem is $O(\frac{1}{\sqrt{t}})$ when $\beta_t = O(\sqrt{t})$.

**Theorem 3.1** *Assume there exists an optimal solution $w^\star$ to the problem (3) with $\Psi(w) = \lambda\|w\|_1$ that satisfies $h(w^\star) \leq D^2$ for some $D > 0$, and let $\phi^\star = \phi(w^\star)$. Let the sequences $\{w_t\}_{t\geq 1}$ be generated by Step 1 in iRDA, and assume $\|g_t\|_* \leq G$ for some constant $G$. Then the expected cost $\mathbf{E}\phi(\bar{w}_t)$ converges to $\phi^\star$ with rate $O(\frac{1}{\sqrt{t}})$*

$$\mathbf{E}\phi(\bar{w}_t) - \phi^\star = O(\frac{1}{\sqrt{t}}),$$

with $\bar{w}_t = \frac{1}{t}\sum_{\tau=1}^{t} w_\tau$. See Appendix A for the proof.

### 3.5    Initialization

To apply iRDA, the weights of a neural network should be initialized differently from that in a normal optimization method such as SGD or its variants. Our initialization is based on LeCun et al. (2012), Glorot and Bengio (2010) and He et al. (2015), with an additional re-scaling. Let $s$ be a scalar, the mean and the standard deviation of the uniform distribution for iRDA is zero and

$$\sigma_{\text{iRDA}} = \sqrt{\frac{s^2}{n}}, \quad n = k^2 c \tag{25}$$

respectively, where $c$ is the number of channels, and $k$ is the spatial filter size of the layer (see He et al. (2015)).

Choosing a suitable $s$ is important when applying iRDA. As shown in Table 5 and Table 6 in Appendix B, if $s$ is too small or too large, the training process could be slowed down and the generalization ability may be affected. Moreover, a small $s$ usually requires much better initial weights, which results in too many samplings in initialization process. In our experiments, a good $s$ for iRDA is usually much larger than $\sqrt{2}$, and unsuitable for SGD algorithms.

### 3.6    Iterative Retraining

Iterative retraining is a method that only updates the non-zero parameters at each iteration. A trained model can be further updated with retraining, thus both the accuracies and sparsity can be improved. See Table 4 for comparisons on CIFAR-10.

---

**Algorithm 1** *The iterative RDA method for $\ell_1$ regularized DNNs*

---

Input:

- A strongly convex function $h(w) = \|w\|_2^2$.
- A nonnegative and nondescreasing sequence $\beta_t = \gamma\sqrt{t}$.

**Step 1: RDA with proper initialization**

Initialize: set $w_0 = 0$, $\bar{g}_0 = 0$ and randomly choose $w_1$ with methods explained in section 3.5.

**for** *t=1,2, ..., T* **do**

Given the sample $z_{i_t}$ and corresponding loss function $f_{i_t}$.

Compute the stochastic gradient

$$g_t = \nabla f_{i_t}(w_t). \tag{26}$$

Update the average gradient:

$$\bar{g}_t = \frac{t-1}{t}\bar{g}_{t-1} + \frac{1}{t}g_t. \tag{27}$$

Compute the next weight vector:

$$w_{t+1} = \arg\min_{w}\left\{\langle\bar{g}_t, w\rangle + \lambda\|w\|_1 + \frac{\beta_t}{2t}\|w\|_2^2\right\}. \tag{28}$$

**Step 2: iterative retraining**

**for** *t=T+1,T+2,T+3, ...* **do**

Given the sample $z_{i_t}$ and corresponding loss function $f_{i_t}$.

Compute the stochastic gradient

$$g_t = \nabla f_{i_t}(w_t). \tag{29}$$

Set $(g_t)_j = 0$ if $(w_t)_j = 0$ for every $j$.

Update the average gradient:

$$\bar{g}_t = \frac{t-1}{t}\bar{g}_{t-1} + \frac{1}{t}g_t. \tag{30}$$

Compute the next weight vector:

$$w_{t+1} = \arg\min_{w}\left\{\langle\bar{g}_t, w\rangle + \lambda\|w\|_1 + \frac{\beta_t}{2t}\|w\|_2^2\right\}. \tag{31}$$

---

## 4 EXPERIMENTS

In this section, $\sigma$ denotes the **sparsity** of a model, i.e.

$$\sigma = \frac{\text{quantity of zero parameters}}{\text{quantity of all parameters}}. \tag{32}$$

All neural networks are trained with mini-batch size 128.

### 4.1 PARAMETERS TEST

We provide a test on different hyper-parameters, so as to give an overview of their effects on performance, as shown in Table 7. We also show that the sparsity and the accuracy can be balanced with iRDA by adjusting the parameters $\lambda$ and $\gamma$, as shown in Table 8. Both tables are put in Appendix C.

### 4.2 MAIN RESULTS

We compare iRDA with several methods including prox-SGD, $\sqrt{t}-$SGD and normal SGD, on different datasets including MNIST, CIFAR-10, CIFAR-100 and ImageNet(ILSVRC2012). The main results are shown in Table 1. Table 2 shows the performance of iRDA on different architectures including ResNet18, VGG16 and VGG19. Table 3 shows the performance of iRDA on different

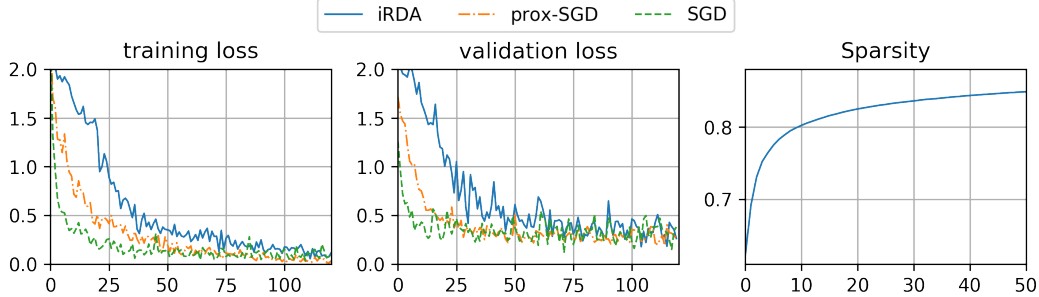

Figure 1: The first 120 epochs of loss curves corresponding to Table 1, and the sparsity curve for another result, where the top-1 validation accuracy is $91.34\%$, and $\sigma = 0.87$.

datasets including MNIST, CIFAR-10, CIFAR-100 and ImageNet(ILSVRC2012). In all tables, SGD denotes stochastic gradient methods with momentum Ruder (2016).

### 4.3 COMPARISON

Currently, many compression methods include human experts involvement. Some methods try to combine other structures in training process to automatize the compression process. For example, He et al. (2018) combines reinforcement learning. iRDA, as an algorithm, requires no extra structure.

As shown above, iRDA can achieve good sparsity while keeping accuracy automatically, with carefully chosen parameters. For CIFAR-10, we compare the performance of iRDA with some other state-of-art compression methods in Table 4. Due to different standards, $\sigma$ is referred to directly or computed from the original papers approximately.

## 5 CONCLUSION

In comparison with many existing rule-based heuristic approaches, the new approach is based on a careful and iterative combination of $\ell_1$ regularization and some specialized training algorithms. We find that the commonly used training algorithms such as SGD methods are not effective. We thus develop iRDA method that can be used to achieve much better sparsity. iRDA is a variant of RDA methods that have been used for some special types of online convex optimization problems in the literature. New elements in the iRDA mainly consist of judicious initialization and iterative retraining. In addition, iRDA method is carefully analyzed on its convergence for convex objective functions.

Many deep neural networks trained by iRDA can achieve good sparsity while keeping the same validation accuracy as those trained by SGD with momentum on many popular datasets. This result shows iRDA is a powerful sparse optimization method for image classification problems in deep learning fields.

Table 1: The main results of different methods. The architecture is ResNet18, and the dataset is CIFAR-10. This table shows the top-1 and top-5 accuracies on the validation dataset. iRDA achieves the highest top-1 accuracy and sparsity. See figure 1 for the corresponding loss curves.

| Method | TOP 1 Acc. | TOP 5 Acc. | $\sigma$ | $\lambda$ | $\gamma$ |
|---|---|---|---|---|---|
| SGD | 92.69 | 99.72 | 0.00 | N/A | N/A |
| prox-SGD | 89.80 | 99.40 | 0.03 | $10^{-5}$ | 0.8 |
| $\sqrt{t}$-prox-SGD | 82.47 | 99.07 | 0.72 | $10^{-8}$ | 1.0 |
| iRDA | **93.47** | 99.69 | **0.95** | $10^{-6}$ | 1.0 |

Table 2: iRDA on different Architectures. This table shows the top-1 and top-5 accuracies on the validation dataset. The results from SGD with momentum are in the brackets. iRDA works well on different CNN architectures.

| ARCHITECTURE | TOP 1 Acc. | TOP 5 Acc. | $\sigma$ | $\lambda$ | $\gamma$ |
|---|---|---|---|---|---|
| ResNet18 | 93.47 (92.69) | 99.69 (99.72) | 0.95 | $10^{-6}$ | 1.0 |
| VGG16 | 93.24 (93.42) | 99.52 (99.79) | 0.94 | $10^{-6}$ | 1.0 |
| VGG19 | 91.87 (91.70) | 99.37 (99.40) | 0.98 | $10^{-5}$ | 1.0 |

Table 3: iRDA on different datasets. The architecture is ResNet18. This table shows the top-1 and top-5 accuracies on the validation dataset. The results from SGD with momentum are in the brackets. iRDA works well on different datasets.

| DATASET | TOP 1 Acc. | TOP 5 Acc. | $\sigma$ | $\lambda$ | $\gamma$ |
|---|---|---|---|---|---|
| MNIST | 99.63 (99.65) | 100.00 | 0.95 | $10^{-6}$ | 0.1 |
| CIFAR-10 | 93.47 (92.69) | 99.69 (99.72) | 0.95 | $10^{-6}$ | 1.0 |
| CIFAR-100 | 72.29 (73.69) | 89.94 (92.43) | 0.56 | $10^{-8}$ | 0.09 |
| ILSVRC2012 | 64.93 (70.58) | 84.92 (89.64) | 0.36 | $10^{-8}$ | 0.1 |

Table 4: iRDA and different state-of-the-art compression methods on CIFAR-10. This table shows the top-1 accuracies on the validation dataset. Due to different standards, $\sigma$ is referred to directly or computed approximately. iRDA achieves almost the same accuracy and sparsity on VGG16, while the sparsity on ResNet18 is much better.

| ARCHITECTURE | METHOD | TOP 1 Acc. | $\sigma$ | $\lambda$ | $\gamma$ |
|---|---|---|---|---|---|
| ResNet-50 | AMC($R_{\text{Param}}$) He et al. (2018) | 93.64 | 0.60 | N/A | N/A |
| ResNet-18 | iRDA | 93.47 | 0.95 | $10^{-6}$ | 1.0 |
| VGG-16 | VIBNet Dai et al. (2018) | 93.80 | 0.94 | N/A | N/A |
| VGG-16 | iRDA | 93.24 | 0.94 | $10^{-6}$ | 1.0 |

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

## A    PROOF OF THEOREM 3.1

One of the differences between RDA Xiao (2010) and iRDA is that the former one takes $w_1 = \arg\min_w h(w)$ whereas the latter one chooses $w_1$ randomly. In the following, we will prove the convergence of iRDA Step 1 for convex problem. The proofs use Lemma 9, Lemma 10, Lemma 11 directly and modify Theorem 1 and Theorem 2 in Xiao (2010). For clarity, we have some general assumptions:

- The regularization term $\Psi(w)$ is a closed convex function with convexity parameter $\sigma$ and dom$\Psi$ is closed.

- For each $t \geq 1$, $f_t(w)$ is convex and subdifferentiable on dom$\Psi$.

- $h(w)$ is strongly convex on dom$\Psi$ and subdifferentiable on rint(dom$\Psi$) and also satisfies

$$w_0 = \arg\min_w h(w) \in \text{Arg}\min_w \Psi(w). \tag{33}$$

  Without loss of generality, assume $h(w)$ has convexity parameter 1 and $\min_w h(w) = 0$.
- There exist a constant $G$ such that

$$\|g_t\|_* \leq G, \ \forall t \geq 1. \tag{34}$$

- Require $\{\beta\}_{t\geq 1}$ be a nonnegative and nondecreasing sequence and

$$\beta_0 = \max\{\sigma, \beta_1\} > 0. \tag{35}$$

  Moreover, we could always choose $\beta_1 \geq \sigma$ such that $\beta_0 = \beta_1$.
- For a random choosing $w_1$, we assume

$$\Psi(w_1) \leq Q. \tag{36}$$

First of all, we define two functions:

$$U_t(s) = \max_{w \in \mathcal{F}_D} \{\langle s, w - w_0 \rangle - t\Psi(w)\}, \tag{37}$$

$$V_t(s) = \max_w \{\langle s, w - w_0 \rangle - t\Psi(w) - \beta_t h(w)\}. \tag{38}$$

The maximum in (37) is always achieved because $\mathcal{F}_D = \{w \in \text{dom}\Psi | h(w) \leq D^2\}$ is a nonempty compact set. Because of (35), we have $\sigma t + \beta_t \geq \beta_0 > 0$ for all $t \geq 0$, which means $t\Psi(w) + \beta_t h(w)$ are all strongly convex, therefore the maximum in (38) is always achieved and unique. As a result, we have dom$U_t = $ dom$V_t = E^*$ for all $t \geq 0$. Moreover, by the assumption (33), both of the functions are nonnegative.

Let $s_t$ denote the sum of the subgradients obtained up to time $t$ in iRDA Step 1, that is

$$s_t = \sum_{\tau=1}^{t} g_\tau = t\bar{g}_t, \tag{39}$$

and $\pi_t(s)$ denotes the unique maximizer in the definition of $V_t(s)$

$$\begin{aligned}
\pi_t(s) &= \arg\max_w \{\langle s, w - w_0 \rangle - t\Psi(w) - \beta_t h(w)\} \\
&= \arg\min_w \{\langle -s, w \rangle + t\Psi(w) + \beta_t h(w)\},
\end{aligned} \tag{40}$$

which then gives

$$w_{t+1} = \pi_t(-s_t). \tag{41}$$

**Lemma A.1** *For any $s \in E^*$ and $t \geq 0$, we have*

$$U_t(s) < V_t(s) + \beta_t D^2. \tag{42}$$

For a proof, see Lemma 9 in Xiao (2010).

**Lemma A.2** *The function $V_t$ is convex and differentiable. Its gradient is given by*

$$\nabla V_t(s) = \pi_t(s) - w_0 \tag{43}$$

*and the gradient Lipschitz continuous with constant $1/(\sigma t + \beta_t)$, that is*

$$\|\nabla V_t(s_1) - \nabla V_t(s_2)\| \leq \frac{1}{\sigma t + \beta_t}\|s_1 - s_2\|_*, \quad \forall s_1, s_2 \in E^*. \tag{44}$$

*Moreover, the following inequality holds:*

$$V_t(s + g) \leq V_t(s) + \langle g, \nabla V_t(s) \rangle + \frac{1}{2(\sigma t + \beta_t)}\|g\|_*^2, \quad \forall s, g \in E^*. \tag{45}$$

The results are from Lemma 10 in Xiao (2010).

**Lemma A.3** *For each $t \geq 1$, we have*

$$V_t(-s_t) + \Psi(w_{t+1}) \leq V_{t-1}(-s_t) + (\beta_{t-1} - \beta_t)h(w_{t+1}). \tag{46}$$

Since $h(w_{t+1}) \geq 0$ and the sequence $\{\beta_t\}_{t \geq 1}$ is nondecreasing, we have

$$V_t(-s_t) + \Psi(w_{t+1}) \leq V_{t-1}(-s_t), \qquad\qquad \forall t \geq 2, \tag{47}$$
$$V_1(-s_1) + \Psi(w_2) \leq V_0(-s_1) + (\beta_0 - \beta_1)h(w_2), \qquad\qquad t = 1. \tag{48}$$

To prove this lemma, we refer to the Lemma 11 in Xiao (2010). What's more, from the assumption 35, we could always choose $\beta_1 \geq \sigma$ such that $\beta_1 = \beta_0$ and

$$V_1(-s_1) + \Psi(w_2) \leq V_0(-s_1), \quad t = 1. \tag{49}$$

The learner's *regret* of online learning is the difference between his cumulative loss and the cumulative loss of the optimal fixed hypothesis, which is defined by

$$R_t(w) = \sum_{\tau=1}^{t}(f_\tau(w_\tau) + \Psi(w_\tau)) - \sum_{\tau=1}^{t}(f_\tau(w) + \Psi(w)), \tag{50}$$

and bounded by

$$\Delta_t = Q + \beta_t D^2 + \frac{G^2}{2}\sum_{\tau=0}^{t-1}\frac{1}{\sigma\tau + \beta_\tau}. \tag{51}$$

**Lemma A.4** *Let the sequence $\{w_t\}_{t \geq 1}$ and $\{g_t\}_{t \geq 1}$ be generated by iRDA Step 1, and assume (34) and (35) hold. Then for any $t \geq 1$ and any $w \in \mathcal{F}_D = \{w \in dom\Psi | h(w) \leq D^2\}$, the regret defined in (50) is bounded by $\Delta_t$*

$$R_t(w) \leq \Delta_t \tag{52}$$

**Proof** First, we define the following *gap* sequence which measures the quality of the solutions $w_1, .., w_t$:

$$\delta_t = \max_{w \in \mathcal{F}_D}\left\{\sum_{\tau=1}^{t}\left(\langle g_\tau, w_\tau - w\rangle + \Psi(w_\tau)\right) - t\Psi(w)\right\}, \quad t = 1, 2, 3, .... \tag{53}$$

and $\delta_t$ is an upper bound on the regret $R_t(w)$ for all $w \in \mathcal{F}_D$, to see this, we use the convexity of $f_t(w)$ in the following:

$$\delta_t \geq \sum_{\tau=1}^{t}\left(f_\tau(w_\tau) - f_\tau(w) + \Psi(w_\tau)\right) - t\Psi(w) = R_t(w). \tag{54}$$

Then, We are going to derive an upper bound on $\delta_t$. For this purpose, we subtract $\sum_{\tau=1}^{t}\langle g_\tau, w_0\rangle$ in (53), which leads to

$$\delta_t = \sum_{\tau=1}^{t}\left(\langle g_\tau, w_\tau - w_0\rangle + \Psi(w_\tau)\right) + \max_{w \in \mathcal{F}_D}\left\{\langle s_t, w_0 - w\rangle - t\Psi(w)\right\}, \tag{55}$$

the maximization term in (55) is in fact $U_t(-s_t)$, therefore, by applying Lemma A.1, we have

$$\delta_t \leq \sum_{\tau=1}^{t} \left( \langle g_\tau, w_\tau - w_0 \rangle + \Psi(w_\tau) \right) + V_t(-s_t) + \beta_t D^2. \tag{56}$$

Next, we show that $\Delta_t$ is an upper bound for the right-hand side of inequality (56). We consider $\tau \geq 2$ and $\tau = 1$ respectively.
For any $\tau \geq 2$, we have

$$V_\tau(-s_\tau) + \Psi(w_{\tau+1}) \leq V_{\tau-1}(-s_{\tau-1}) + \langle -g_\tau, w_\tau - w_0 \rangle + \frac{\|g_\tau\|_*^2}{2(\sigma(\tau-1) + \beta_{\tau-1})},$$

where (47),(39),(45) and (43) are used. Therefore, we have

$$\langle g_\tau, w_\tau - w_0 \rangle + \Psi(w_{\tau+1}) \leq V_{\tau-1}(-s_{\tau-1}) - V_\tau(-s_\tau) + \frac{\|g_\tau\|_*^2}{2(\sigma(\tau-1) + \beta_{\tau-1})}, \quad \forall \tau \geq 2.$$

For $\tau = 1$, we have a similar inequality by using (49)

$$\langle g_1, w_1 - w_0 \rangle + \Psi(w_2) \leq V_0(-s_0) - V_1(-s_1) + \frac{\|g_1\|_*^2}{2\beta_0}.$$

Summing the above inequalities for $\tau = 1, ..., t$ and noting that $V_0(-s_0) = V_0 = 0$, we arrive at

$$\sum_{\tau=1}^{t} \left( \langle g_\tau, w_\tau - w_0 \rangle + \Psi(w_{\tau+1}) \right) + V_t(-s_t) \leq \sum_{\tau=1}^{t} \frac{\|g_\tau\|_*^2}{2(\sigma(\tau-1) + \beta_{\tau-1})}.$$

Since $\Psi(w_{t+1}) \geq 0$, we subtract it from the left hand side and add $\Psi(w_1)$ to both sides of the above inequality yields

$$\sum_{\tau=1}^{t} \left( \langle g_\tau, w_\tau - w_0 \rangle + \Psi(w_\tau) \right) + V_t(-s_t) \leq \Psi(w_1) + \frac{1}{2} \sum_{\tau=1}^{t} \frac{\|g_\tau\|_*^2}{2(\sigma(\tau-1) + \beta_{\tau-1})}. \tag{57}$$

Combing (54), (56), (57) and using assumption (34) and (36) we conclude

$$R_t(w) \leq \delta_t \leq \Delta_t = Q + \beta_t D^2 + \frac{G^2}{2} \sum_{\tau=0}^{t-1} \frac{1}{\sigma\tau + \beta_\tau}.$$

**Lemma A.5** *Assume there exists an optimal solution $w^\star$ to the problem (3) that satisfies $h(w^\star) \leq D^2$ for some $D > 0$, and let $\phi^\star = \phi(w^\star)$. Let the sequences $\{w_t\}_{t \geq 1}$ be generated by iRDA Step 1, and assume $\|g_t\|_* \leq G$ for some constant $G$. Then for any $t \geq 1$, the expected cost associated with the random variable $\bar{w}_t$ is bounded as*

$$\mathbf{E}\phi(\bar{w}_t) - \phi^\star \leq \frac{1}{t}\Delta_t.$$

**Proof** First, from the definition (50), we have the regret at $w^\star$

$$R_t(w^\star) = \sum_{\tau=1}^{t} (f(w_\tau, z_\tau) + \Psi(w_\tau)) - \sum_{\tau=1}^{t} (f(w^\star, z_\tau) + \Psi(w^\star)),$$

Let $\mathbf{z}[t]$ denote the collection of i.i.d. random variables $(z, ..., z_t)$. We note that the random variable $w_\tau$, where $1 \leq w \geq t$, is a function of $(z_1, ..., z_{\tau-1})$ and is independent of $(z_\tau, ..., z_t)$. Therefore

$$\mathbf{E}_{\mathbf{z}[t]} \left( f(w_\tau, z_\tau) + \Psi(w_\tau) \right) = \mathbf{E}_{z[\tau-1]} \left( \mathbf{E}_\tau f(w_\tau, z_\tau) + \Psi(w_\tau) \right) = \mathbf{E}_{z[\tau-1]}\phi(w_\tau) = \mathbf{E}_{z[t]}\phi(w_\tau),$$

and

$$\mathbf{E}_{\mathbf{z}[t]} \left( f(w^\star, z_\tau) + \Psi(w^\star) \right) = \mathbf{E}_\tau f(w^\star, z_\tau) + \Psi(w^\star) = \phi(w^\star) = \phi^\star.$$

Since $\phi^\star = \phi(w^\star) = \min_w \phi(w)$, we have the expected regret

$$\mathbf{E}_{\mathbf{z}[t]} R_t(w^\star) = \sum_{\tau=1}^{t} \mathbf{E}_{\mathbf{z}[t]}\phi(w_\tau) - t\phi^\star \geq 0. \tag{58}$$

Then, by convexity of $\phi$, we have

$$\phi(\bar{w}_t) = \phi\left(\frac{1}{t}\sum_{\tau=1}^{t} w_\tau\right) \leq \frac{1}{t}\sum_{\tau=1}^{t}\phi(w_\tau). \tag{59}$$

Finally, from (59) and (58), we have

$$\mathbf{E}_{\mathbf{z}[t]}\phi(\bar{w}_t) - \phi^\star \leq \frac{1}{t}\left(\sum_{\tau=1}^{t}\mathbf{E}_{\mathbf{z}[t]}\phi(w_\tau) - t\phi^\star\right) = \frac{1}{t}\mathbf{E}_{\mathbf{z}[t]}R_t(w^\star).$$

Then the desired follows from that of Lemma A.4.

**Proof of Theorem 3.1** From Lemma A.5, the expected cost associated with the random variable $\bar{w}_t$ is bounded as

$$\mathbf{E}\phi(\bar{w}_t) - \phi^\star \leq \frac{1}{t}\left(Q + \beta_t D^2 + \frac{G^2}{2}\sum_{\tau=0}^{t-1}\frac{1}{\sigma\tau + \beta_\tau}\right), \tag{60}$$

Here, we consider $\ell_1$ regularization function $\Psi(w) = \lambda\|w\|_1$ and it is a convex but not strongly convex function, which means $\sigma = 0$. Now, we consider how to choose $\beta_t$ for $t \geq 1$ and $\beta_0 = \beta_1$. First if $\beta_t = \gamma t$, we have $\frac{1}{t}\cdot\gamma t D^2 = \gamma D^2$, which means the expected cost does not converge. Then assume $\beta_t = \gamma t^\alpha$, $\alpha > 0$ and $\alpha \neq 1$, the right hand side of the inequality (60) becomes

$$\frac{1}{t}\left(Q + \gamma D^2 t^\alpha + \frac{G^2}{2\gamma}\sum_{\tau=0}^{t-1}\frac{1}{\tau^\alpha}\right) \leq \frac{1}{t}\left[Q + \gamma D^2 t^\alpha + \frac{G^2}{2\gamma}\left(2 + \sum_{\tau=2}^{t-1}\frac{1}{\tau^\alpha}\right)\right]$$

$$\leq \frac{1}{t}\left[Q + \gamma D^2 t^\alpha + \frac{G^2}{2\gamma}\left(2 + \int_1^{t-1}\frac{1}{\tau^\alpha}\right)\right] \sim O(t^{\alpha-1} + t^{-\alpha}).$$

From above, we see that if $0 < \alpha < 1$, the expected cost converges and the optimal convergence rate $O(t^{-\frac{1}{2}})$ achieves when $\alpha = \frac{1}{2}$. Then we proved the Theorem 3.1.

## B INITIALIZATION

Table 5: Different initialization scalars on CIFAR-10 with iRDA. The architecture is ResNet18. $\lambda = 10^{-6}$ and $\gamma = 1.0$. This table shows the top-1 accuracies on the validation dataset. All models are trained for 120 epochs.

| $s$ | TOP 1 Acc. | TOP 5 Acc. | $\sigma$ |
|---|---|---|---|
| 1, 2 | 10.00 | 50.00 | N/A |
| 3 | 85.52 | 99.24 | 0.98 |
| 4 | 86.72 | 99.45 | 0.97 |
| 5 | 90.03 | 99.44 | 0.95 |
| 10 | 90.67 | 89.50 | 0.94 |
| 100 | 91.41 | 99.58 | 0.84 |
| 1000 | 90.36 | 99.62 | 0.63 |
| 10000 | 71.80 | 97.94 | 0.34 |
| 20000 | 68.06 | 97.39 | 0.99 |

Table 6: Different initialization scalars on CIFAR-100 with iRDA. The architecture is ResNet18. $\lambda = 10^{-8}$ and $\gamma = 0.1$. This table shows the top-1 accuracies on the validation dataset. All models are trained for 120 epochs.

| $s$ | TOP 1 Acc. | TOP 5 Acc. | $\sigma$ |
|---|---|---|---|
| 1 | 63.67 | 87.85 | 0.91 |
| 2 | 66.90 | 88.53 | 0.60 |
| 5 | 65.47 | 88.09 | 0.60 |
| 10 | 65.54 | 88.21 | 0.42 |
| 15 | 64.22 | 87.53 | 0.43 |
| 25 | 63.06 | 88.10 | 0.50 |
| 30 | 62.75 | 86.80 | 0.42 |
| 50 | 64.48 | 87.14 | 0.38 |
| 100 | 60.00 | 86.14 | 0.36 |

## C  PARAMETERS TEST

Table 7: Fix $\gamma = 1.0$ and test different $\lambda$ with different methods on CIFAR-10. The architecture is ResNet18. All models are trained for 120 epochs. This table shows the top-1 accuracies on the validation dataset. We have shown why prox-SGD will give poor sparsity, and although $\sqrt{t}$-prox-SGD may introduce greater sparsity, it is not convergent. Finally, iRDA gives the best result, on both the top-1 accuracy and the sparsity.

| $\lambda$ | $10^{-2}$ | $10^{-3}$ | $10^{-4}$ | $10^{-5}$ | $10^{-6}$ | $10^{-7}$ | $10^{-8}$ |
|---|---|---|---|---|---|---|---|
| iRDA(without Retraining) | 10.00 | 38.73 | 74.04 | 86.39 | **90.67** | 90.14 | 88.38 |
| $\sigma$ | 1.00 | 1.00 | 1.00 | 0.99 | 0.94 | 0.61 | 0.16 |
| $\sqrt{t}$-prox-SGD | 10.00 | 10.00 | 10.00 | 10.00 | 8.43 | 49.20 | 82.47 |
| $\sigma$ | 1.00 | 1.00 | 1.00 | 1.00 | 1.00 | 0.99 | 0.72 |
| prox-SGD | 17.90 | 75.11 | 88.74 | 88.20 | 89.02 | 86.92 | 87.75 |
| $\sigma$ | 1.00 | 0.97 | 0.52 | 0.01 | 0.00 | 0.00 | 0.00 |

Table 8: Different $\lambda$ and $\gamma$ with iRDA on CIFAR-10. The architecture is ResNet18. All models are trained for 120 epochs. This table shows the top-1 accuracies on the validation dataset. The highest accuracy is 92.07% with sparsity 0.84, and with a lower accuracy 90.67% we get a sparsity of 0.94. The result shows that we can balance the accuracy and the sparsity with iRDA, by adjusting the parameters $\gamma$ and $\lambda$.

| $\gamma$ | 0.1 | 0.2 | 0.3 | 0.4 | 0.5 | 0.6 | 0.7 | 0.8 | 0.9 | 1.0 |
|---|---|---|---|---|---|---|---|---|---|---|
| $\lambda = 10^{-6}$ | 90.73 | 91.68 | 91.88 | **92.07** | 91.68 | 90.33 | 90.88 | 89.40 | 89.72 | 90.67 |
| $\sigma$ | 0.78 | 0.81 | 0.82 | 0.84 | 0.89 | 0.83 | 0.92 | 0.94 | 0.94 | **0.94** |
| $\lambda = 10^{-7}$ | 90.87 | 91.39 | 91.88 | 90.77 | 91.80 | 90.46 | 90.04 | N/A | 90.19 | 90.14 |
| $\sigma$ | 0.41 | 0.45 | 0.49 | 0.55 | 0.53 | 0.55 | 0.60 | N/A | 0.61 | 0.61 |
| $\lambda = 10^{-8}$ | 91.42 | 91.00 | 91.62 | 92.25 | 91.06 | 89.71 | 87.11 | 90.45 | 86.31 | 88.38 |
| $\sigma$ | 0.09 | 0.12 | 0.14 | 0.12 | 0.13 | 0.14 | 0.13 | 0.15 | 0.16 | 0.16 |

