# OpenReview forum: "iRDA Method for Sparse Convolutional Neural Networks"
_ICLR.cc/2019/Conference_

### Official Review · AnonReviewer2 · 2018-11-03
**Not enough novelty**

**Rating:** 3
**Confidence:** 5

**Review:**

This paper claims to propose a new iRDA method. Essentially, it is just dual averaging with \ell_1 penalty and an \ell_2 proximal term. The O(1/\sqrt{t}) rate is standard in literature. This is a clear rejection.

---

### Official Review · AnonReviewer3 · 2018-11-05
**Algorithm and presentation are flawed**

**Rating:** 3
**Confidence:** 4

**Review:**

The submission made a few modifications to the RDA (regularized dual averaging) optimization solver to form the proposed "iterative RDA (iRDA)" algorithm, and shows that empirically the proposed algorithm could  reduce the number of non-zero parameters by an order of magnitude on CIFAR10 for a number of benchmark network architectures (Resnet18, VGG16, VGG19).

The experimental result of the paper is strong but the algorithm and also a couple of statements seem flawed. In particular:

* For Algorithm 1, consider the case when lamda=0 and t -> infinity,  the minimization eq (28) goes to negative infinity for any non-zero gradient, which corresponds to an update of infinitely large step size. It seems something is wrong.

* Why in Step 2 the algorithm sets both g_t and w_t to 0 during each iterate? It looks so wrong.

*The whole paper did not mention batch size even once. Does the algorithm apply only with batch size=1?

*What is the "MRDA" method in the figure? Is it mentioned anywhere in the paper?

*What are "k", "c"  in eq (25)? Are they defined anywhere in the paper?

*Theorem states 1/sqrt(t) convergence but eq (28), (31) have updates of unbounded step size. How is this possible?

---

### Official Review · AnonReviewer1 · 2018-11-13
**The paper is not written well and needs major modifications. The contribution of the paper which is analyzing RDA with arbitrary init point is a small incremental contribution.**

**Rating:** 3
**Confidence:** 5

**Review:**

iRDA Method for sparse convolutional neural networks

This paper considers the problem of training a sparse neural network. The main motivation is that usually all state of the art neural network’s size or the number of weights is enormous and saving them in memory is costly. So it would be of great interest to train a sparse neural network. To do so, this paper proposed adding l1 regularizer to RDA method in order to encourage sparsity throughout training. Furthermore, they add an extra phase to  RAD algorithm where they set the stochastic gradient of zero weights to be zero. They show experimentally that the method could give up to 95% sparsity while keeping the accuracy at an acceptable level.
More detail comments:

1- In your analysis for the convergence, you totally ignored the second step. How do you show that with the second step still the method converge?

2- \bar{w} which is used in the thm 1, is not introduced.

3- In eq 5, you say g_t is subfunction. What is it?

4- When does the algorithm switch from step 1 to step 2?

5- In eq 35 what is \sigma?

6- What is the relation between eq 23 and 24? The paper says 23 is an approximation for 24 but the result of 23 is a point and 24 is a function.

7- What is MRDA in the Fig 1?

---

### Meta-Review · Area_Chair1 · 2018-12-15
**Limited novelty.**

**Confidence:** 5
**Recommendation:** Reject

**Metareview:**

This paper proposes an “iterative” regularized dual averaging method to sparsify CNN weights during learning. The main contribution seems to be in an iterative procedure where the weights are pruned out greedily by observing the sparsity of the averaged gradients. The reviewers agree that the idea seems straightforward and novelty is limited. For this reason, I recommend to reject this paper.